# Improved Understanding of Interactions between Risk Factors for Child Obesity May Lead to Better Designed Prevention Policies and Programs in Indonesia

**DOI:** 10.3390/nu12010175

**Published:** 2020-01-08

**Authors:** Hamam Hadi, Esti Nurwanti, Joel Gittelsohn, Andi Imam Arundhana, Dewi Astiti, Keith P. West, Michael J. Dibley

**Affiliations:** 1Department of Nutrition, Faculty of Health Science, Universitas Alma Ata, Yogyakarta 55183, Indonesia; estinurwanti@uaa.ac.id (E.N.); dewiastiti@uaa.ac.id (D.A.); 2Alma Ata Center for Healthy Life and Foods (ACHEAF) Universitas Alma Ata, Yogyakarta 55183, Indonesia; 3Center for Human Nutrition, Department of International Health, Bloomberg School of Public Health, The Johns Hopkins University, Baltimore, MD 21205, USA; jgittel1@jhu.edu (J.G.);; 4Department of Nutrition, Faculty of Public Health, Hasanuddin University, Makassar 90245, Indonesia; andi.imam@unhas.ac.id; 5Central Clinical School, Faculty of Medicine and Health Science, The University of Sydney, Sydney 2050, Australia; 6Global Public Health Nutrition, Sydney School of Public Health, Sydney Medical School, The University of Sydney, Sydney 2006, Australia; michael.dibley@sydney.edu.au

**Keywords:** sedentary behaviors, fruit and vegetable, junk food, school children, Indonesia, obesity

## Abstract

The nutrition transition in low-middle income countries is marked by rising intakes of highly caloric, low nutrient-dense (junk) foods, decreasing intakes of fruits and vegetables, and sedentary behavior. The objective of this study was to explore interactions among fruit-and-vegetable intake, junk food energy intake, sedentary behavior, and obesity in Indonesian children. We conducted this school-based, case-control study in 2013 in Yogyakarta Special Province, Indonesia. The cases were 244 obese children aged 7–12 years having a BMI ≥95th percentile of an age- and sex-specific distribution from the Centers for Disease Control and Prevention. The controls (*n* = 244) were classroom-matched children with a BMI <85th percentile. Using conditional logistic regression, the relative odds (95% confidence intervals; OR: 95% CI) of obesity given reported frequent fruit-and-vegetable intake (≥3 servings/day), low junk food energy (≤1050 kcal/day) intake and low sedentary behavior (<5 h/day) was 0.46 (0.30–0.69), 0.61 (0.37–0.98), and 0.18 (0.12–0.28), respectively. Effect sizes were dose-responsive and appeared additive. For example, children with low sedentary behavior and frequent fruit-and-vegetable intake were 92% less likely (OR = 0.08; 0.04–0.15) to be obese than children not exceeding either of these thresholds. Similarly, children frequently eating fruits and vegetables and reporting a low junk food energy intake were 70% less likely (OR = 0.30; 0.15–0.59) to be obese. The findings were unchanged after adjusting for child, maternal, and household covariates. Preventive interventions for child obesity need multiple components to improve diets and raise levels of physical activity rather than just addressing one of the three types of assessed behaviors.

## 1. Introduction

Childhood obesity has become a major public health problem worldwide, with its prevalence increasing dramatically over the last two decades [1]. Globally, the number of school-age children and adolescents with obesity has risen more than 10-fold, from 11 million to 124 million in the last 40 years [2]. Obese children are more likely to stay obese in adulthood than non-obese children and are more likely to experience non-communicable diseases in the future [3]. In developing countries, the prevalence of childhood obesity varies from 5% to 40%, warranting increasing concern [4]. In Indonesia, the prevalence of obesity in school-age children (5–12 years) has risen from 8.0% in 2013 [5] to 9.2% in 2018 [6].

Excessive energy intakes relative to requirements can lead to weight gain, usually reflecting an unhealthy diet high in fat and sugar, inadequate in fruits and vegetables, and thus low in fiber, coupled with a physically inactive lifestyle [4,7,8,9]. In Indonesia, there are reports that 93.5% of children aged ten years and older consumed fruits and vegetables less frequently than recommended by the Ministry of Health [5]. Further, 53% of these children consumed sweet food or beverages, and 41% fatty foods, more than once per day [5]. Besides, in Indonesia, nearly a quarter of school-aged children are estimated to be sedentary in their behavior (i.e., being inactive for ≥6 daytime/wakeful hours/day) [5]. Thus, the exposure of Indonesian children to major causal, although alterable, practices directly increases their risk of becoming obese or incurring obesity-related health problems in their lives [10,11].

Fruit and vegetable intakes are inversely associated with overweight and obesity [12,13,14]. We have previously reported that obese school-aged children in Indonesia had a lower fruit and vegetable intake than non-obese children [15]. A systematic review suggested that adequate fruit and vegetable intakes may lower the risk of becoming obese [14]. Fruits and vegetables are high in fiber, which increases satiety, and result in fewer calories consumed [16,17]. Fruits and vegetables also have a lower glycemic index (GI)/glycemic load, which can reduce food energy intake and body weight [18]. The flavonoids contained in fruits and vegetables can also improve insulin sensitivity [19], assisting in the regulation of energy balance [20]. However, the flavonoid content in fruits and vegetables is influenced by many factors including fertilizers [21], ripening stage of fruit [22], storage method [23], and cooking method [24]. We have reported that Indonesian children had higher exposure to junk food advertisements than their similar-aged peers in other Asia-Pacific countries [25]. The obese children from this study were exposed more frequently to junk food advertisements. This exposure was also associated with significantly higher junk food energy intake in obese than non-obese children [26]. Other studies have shown that children with higher junk food consumption are at a higher risk for obesity [27,28,29].

Studies of risk factors for childhood obesity often examine persistent sedentary behavior, excessive junk food intakes, and low fruit and vegetable consumption separately—this approach limits relative comparisons or the quantification of the effects of combined risk factors. Additionally, only a few studies have investigated the interaction between low fruit and vegetable intakes, high junk food energy intakes, and sedentary behavior in evaluating their combined effects on the risk of obesity. There are no studies from Indonesia that have examined the effects of combinations of these risk factors on child obesity.

Improved understanding of these interactions could lead to better-designed policies and programs for prevention. Therefore, this study examined fruit and vegetable intake, junk food energy intake, and sedentary behavior as risk factors for obesity in Indonesian children, as well as the effects of combinations of these risk factors. We hypothesized that among Indonesian schoolchildren, low consumption of fruit and vegetables combined with either high sedentary behavior or high junk food energy intake is associated with a higher risk of obesity, compared to their individual effects.

## 2. Materials and Methods

### 2.1. Study Population and Survey

We derived the study sample from a cross-sectional school-based population survey measuring the nutritional status of the children aged 6 to 12 years in Bantul Regency and Yogyakarta City, representing both rural and urban settings in Indonesia. We conducted the study in 2013, and we obtained the primary school data for research sites from the Education Office in the two study areas. We combined the student’s data from each school in Yogyakarta and Bantul to create a database of all students (*n* = 119,338). We used the probability proportional to size (PPS) sampling method [30], to select 30 primary schools out of 362 primary schools in Bantul Regency and another 30 primary schools out of 167 primary schools in the city of Yogyakarta. We then screened 3483 children from 60 schools, and trained undergraduate students from the school of nutrition, Alma Ata University, measured their anthropometric status. Of 3483 children screened in the survey, 580 (16.65%) were obese, and the other 2903 children (83.35%) were not obese. We also collected data on sex, age, address of each selected school, and the household ownership of TVs, motorbikes, and cars reported by each child. We did not collect data on sedentary behavior, fruit and vegetable intakes, and junk food intakes during the screening survey.

### 2.2. Case-Control Selection

A child was defined as obese if his or her Body Mass Index (BMI) was ≥95th percentile of the age- and sex-specific body mass index proposed by the Centers for Disease Control and Prevention (Atlanta, GA, USA). We did not obtain any cases or controls from the 6th grades because the schools did not permit further interviews as the students had to focus on national school examinations. We randomly selected 244 obese children from the 1st to 5th grades. We paired each case with one control child (*n* = 244) who sat closest and to the right of a case in the same classroom and whose BMI was below the 85th percentile. We interviewed the cases and controls on the same day. This sample size had sufficient power to examine sedentary behavior, fruit and vegetable intake, and junk food energy intake individually as risk factors for childhood obesity and their interactions as potential effect modifiers [31]. The survey team administered detailed structured questionnaires to cases and controls, as well as their parents, to collect data on sedentary behavior, fruit and vegetable intake, and junk food intakes. The design of this study has been reported elsewhere [15,26,32].

### 2.3. Measurements

In addition to anthropometric status, we measured sociodemographic characteristics (sex, age, mother’s education, mother’s job, father’s education, father’s job, total household expenditures monthly) from each case and control using structured questionnaires. We calculated the obesity status using the WHO Anthro plus software [33]. We measured the fruit and vegetable intakes using a semi-quantitative food frequency (SQFF), which we developed for the study population [34,35]. It contained a food list of 109 junk foods and 135 non-junk foods. We collected FFQs by asking the cases and controls how many times they ate a beef burger, pizza, spaghetti, and other junk foods per day, per week, per month, or per three months. Responses were cross-checked with mothers about intake frequency in the last three months. Energy intakes were calculated by multiplying the approximate weight of a food × frequency (in a day, week, and month) × energy content of the food, and converted into intake per day. We calculated the energy content of the food from nutrient contents/facts labels. Food models were used to help determine portions and weights consumed. We used a modified child physical activity questionnaire (M-CPAQ) to record sedentary activities of the children in the last seven (7) days [36,37], excluding time for sleeping [37]. We invited the children to recall all activities from the time they woke up in the morning until they went to sleep at night, including what they did before, during, and after school. We have included the SQFF (Appendix A) and M-CPAQ (Appendix A) used in this study in the Appendix A.

### 2.4. Variable Categories

To examine fruit and vegetable intake, junk food energy intake, and sedentary behavior as potential effect modifiers, we treated each as a dichotomous variable; low fruit and vegetable intake (<3 servings/d) or more Frequent fruit and vegetable intake (≥3 servings/d); low junk food energy intake (<1050 kcal/d) or high junk food energy intake (≥1050 kcal/d); low sedentary behavior (<5 h/d; i.e., active) or high sedentary behavior (≥5 h/d; inactive). We examined the dose-response pattern of the relationship between fruit and vegetable intake, junk food energy intake, sedentary behavior, and child obesity. We treated fruit and vegetable intake, junk food energy intake, and sedentary behavior each as a categorical variable. We categorized the fruit and vegetable intake as never, one serving/d, two serving/d, and ≥three serving/d. We categorized the junk food energy intake as low junk food energy (<700 kcal/d), moderate junk food energy (≥700 to 1049 kcal/d), and high junk food energy intake (≥1050 kcal/d), and the sedentary behavior as low sedentary behavior (<3.75 h/d), moderate sedentary behavior (3.75–4.99 h/d), and high sedentary behavior (≥5 h/d).

### 2.5. Statistical Analysis

We initially performed Chi-square or *t*-test analysis to assess significant differences between cases and controls. We computed the Odds Ratios (OR) with 95% confidence interval (95% CI) for sedentary activity, fruit-and-vegetable consumption (as a single food group), and junk food consumption variables using simple logistic regression. We considered a potential risk factor protective if the OR was below 1.00 and the confidence intervals did not transect 1.00, and potentially harmful if the OR was above 1.00 and the confidence intervals did not transect 1.00. We performed conditional multiple logistic regression analyses by adjusting for calorie intake, demographic, and socio-economic factors. We performed all data analysis using STATA v.15 MP (StataCorp LLC, TX, USA).

### 2.6. Ethical Approvals

Parents agreed to the participation of their children using an informed consent form. The aim of the study and how the participants would be involved in this study was described to the parents before collecting the data. The study obtained ethical approval from the Ethics Committee of Faculty of Medicine, Gadjah Mada University.

## 3. Results

Obese and non-obese children were comparable in terms of age, mothers’ ages, education and job, fathers’ ages and education, household expenditure, and TV ownership. However, obese children were more likely to be male, sedentary, eat fewer servings of fruits and vegetables, and have higher junk food intake (Table 1).

### 3.1. Dose-Response Related to Obesity Risk Behaviors

As a dichotomous analysis, we observed ORs (95% CI) for obesity of 0.46 (0.30–0.69), 0.61 (0.37–0.98), and 0.18 (0.12–0.28) associated with more frequent fruit and vegetable intake (≥3 serving/d), low junk food energy intake (<1050 kcal/d) and low sedentary behavior in compared to less frequent fruit and vegetable intake, more frequent junk food energy intake, and more frequent sedentary behavior, respectively (Table 2). We found a dose-response decrease in the relative odds of obesity with increasing servings of fruit and vegetables. In contrast, we found a dose-response increase in relative odds of obesity with increasing sedentary behavior and junk food energy intake. The ORs associated with fruit and vegetable intake, sedentary behavior, and high junk food energy intake remained virtually the same after adjusting for sex, age of the child, residence, mother’s education, household expenditure, and TV ownership (Table 2).

### 3.2. Interactions between FV Intake and Sedentary Behavior

Dichotomizing fruit and vegetable intake, we observed that a low (<3 servings/d) vs. high (≥3 servings/d) daily frequency was sufficient to distinguish obese from non-obese. Children with high fruit and vegetable intakes were 54% (OR = 0.46; 95% CI = 0.30–0.69) less likely to be obese than those with low fruit and vegetable intakes. Similar results were found with low sedentary behavior (<5 h/d) vs. high sedentary behavior (≥5 h/d). Children with low sedentary behavior were 82% (OR = 0.18; 95% CI = 0.12–0.28) less likely to be obese than those with high sedentary behavior (Table 2). We found that combined effects remained statistically significant after adjusting for sex, age of the child, mother’s education, household expenditure, TV ownership, and calorie intakes (Figure 1). Compared to sedentary children with low fruit and vegetable intakes, sedentary children with higher fruit and vegetable intakes, non-sedentary children with low fruit and vegetable intakes, and non-sedentary children with higher intakes of fruit and vegetable were 66%, 18%, and 8% as likely to be obese (Figure 1), respectively.

### 3.3. Interaction between FV Intake and Junk Food Energy Intake

In this analysis, we treated junk food energy intake as a dichotomous variable. We classified junk food energy intakes as low if it was <1050 kcal/d, and high junk food energy intake if it was ≥1050 kcal/d. We used a cutoff point of 1050 kcal/d as the threshold for two reasons. First, the average junk food energy intake in non-obese children was about 700 kcal/d. Thus, the threshold had to be above the junk food energy intake for non-obese children to ensure that it would have a strong discriminating power. Second, according to Indonesia Dietary Allowance (RDA) [38], the RDA of energy intake for children ages 6–12 years old is about 2100 kcal/d with a little difference between boys and girls after ten years of age. Thus, a cutoff point of 1050 kcal/d would represent about half of the RDA for children in this age group.

Compared to children with both low fruit and vegetable and high junk food energy intakes, children with low fruit and vegetable and low junk food energy intakes, children with higher fruit and vegetable and also higher junk food energy intakes, and children with higher fruit and vegetable and low junk food energy intakes were 74%, 44%, and 30% as likely to be obese (Figure 2), respectively.

## 4. Discussion

This study is one of the first to examine the interaction between fruit and vegetable intake, sedentary behavior, and junk food energy intakes and childhood obesity in a low-middle income country. We found the risk of school-aged childhood obesity to be inversely associated with the reported frequency of fruit and vegetable intake, irrespective of junk food intake and level of daily activity. These findings are in line with previous studies reporting that fruit and vegetable consumption was associated with lower BMI [39,40], reduced weight gain [13], or adiposity [41,42]. We found a strong additive interaction between fruit and vegetable intake and sedentary behavior. Children with low sedentary behavior were less likely to be obese, especially when they had high fruit and vegetable intake. We also found a strong interaction between fruit and vegetable intake and junk food energy intake. Children with low junk food energy intake were less likely to be obese, especially when they had high fruit and vegetable intakes. The interactions remained statistically significant after adjusting for sex, age of the child, residence, mother’s education, household expenditure, and TV ownership.

We also found that usual daily hours of sedentary behavior were also associated with an increased risk of child obesity, independent of dietary intake. The more hours spent each day doing sedentary activities, the more likely children would be obese. These findings align with previous studies in children in the UK and the US that reported a positive association between the extent of sedentary behavior and obesity in children [43,44].

Accordingly, frequent junk food intake expressed as kilocalories per day from foods such as burgers, fried chicken, and French fries was also strongly associated with the risk of obesity, independent of other aspects of diet and activity. Previous studies have reported the bivariate association but not its independent contribution to risk, regardless of other aspects of diet and activity in school-aged children in societies undergoing the “nutrition transition” [8,45,46,47], as is rapidly occurring in Indonesia [48,49,50].

We also found additive effects for the risks of low fruit and vegetable plus high junk food energy intakes, and low fruit and vegetable and sedentary activity, on the risk of childhood obesity. There has been no previous study exploring interactive effects between fruit and vegetable and junk food intakes on the risk of childhood obesity in a low-middle income country. These novel findings could help the Indonesian government and policymakers in the region and globally in designing, targeting, and evaluating child obesity prevention programs. Based on data of the Indonesian Basic National Health Research conducted in 2013, we found that the prevalence of school children under 13 years of age with or exposed to high fruit and vegetable intake (≥3 serving/d) and low sedentary behavior (<5 h/d) was 22.6%, the prevalence of school children with low fruit and vegetable intake (<3 serving/d) and low sedentary behavior was 40.9%, while the prevalence of school children with high fruit and vegetable intake but also high sedentary behavior (≥5 h/d) was 13.5% [51]. Accordingly, based on the prevalence and the adjusted ORs presented in Figure 1, we used Levin’s formula to calculate the population attributable risk [52]. Our results suggest a potential reduction of up to 72.2% in child obesity prevalence if prevention programs successfully increased fruit and vegetable intake and reduced the sedentary time of school children. They also suggest a potential reduction of up to 65.1% in child obesity if prevention programs only increased daily physical activity, but did not increase fruit and vegetable intake. Alternatively, estimates from this study suggest that the expected reduction in child obesity might only be 6.5% from current levels in Indonesia if policy efforts successfully but solely focused on improving dietary vegetable and fruit intakes without addressing physical inactivity.

Eating junk food, such as burgers, fried chicken, and French fries, has been a growing part of the diet in Indonesia during the last two decades [53]. Junk foods eaten by Indonesian children are not only from western fast foods but also include local fast foods, and contribute to increased child obesity risk [28,54]. Our study shows that children who have less than 50% of their total energy requirement from junk food were 39% less likely to be obese than their peers with higher caloric intake from junk foods. More importantly, Indonesian children with low fruit and vegetable consumption and low junk food intake are 1.4 times less likely to be obese than children with low fruit and vegetable consumption, but high junk food energy intake. Similarly, children with high fruit and vegetable consumption and high junk food energy intake are 2.3 times less likely to be obese than children with low fruit and vegetable consumption but high junk food energy intake. Similarly, children with high fruit and vegetable consumption and low junk food energy intake are 3.3 times less likely to be obese than children with low fruit and vegetable consumption and high junk food energy intake.

In Indonesia, normally local junk foods are provided to children separately from fruits or vegetables and are rarely sold in combination with these foods. However, there are combinations of different types of salads sold with some Western fast foods [55], and this is a possible solution for the Indonesian context. Our findings suggest the need to combine fruit and vegetables with other foods and health education to increase the sale of these healthy foods [56]. This approach might help prevent Indonesian children from becoming obese.

A strength of this study is its large sample size. It allowed us to look at the individual and combined effects of multiple risk factors on childhood obesity. The questionnaires about fruit and vegetable consumption, physical activity, and junk food energy intake were specific, and the additional portion size and physical activity cards we used potentially led to reduced reporting bias from study participants. Second, this analysis is the first in Indonesia, examining the combined effects of fruit and vegetable intakes, sedentary behavior, and junk food energy intake on child obesity.

The case-control design is a limitation of this study as it restricts our ability to assign causation. A preferable design would be a cohort study to test the individual and the combined effects of fruit and vegetable intake, sedentary behavior, and junk food energy intake on obesity in Indonesian school children by taking socio-demographic status and different environments into account.

## 5. Conclusions

The findings of this study suggest that the combined effect of fruits and vegetable intake, sedentary behavior, and junk food energy intake on obesity in Indonesian school children is much stronger than their individual effects. Future intervention trials should consider testing combined intervention strategies (e.g., improving diet and reducing sedentary behavior) on obesity among Indonesian school children. Multiple interventions that address dietary and physical activity improvement in combination are more likely to be effective in preventing childhood obesity in the Indonesian context.

## Figures and Tables

**Figure 1 nutrients-12-00175-f001:**
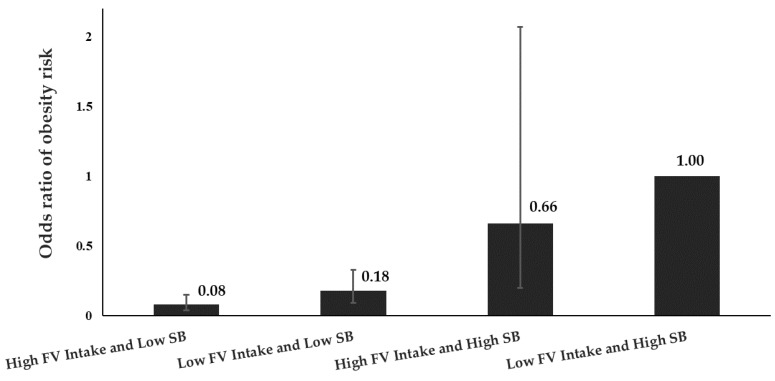
The interaction between daily fruit and vegetable intake and sedentary behavior. We generated the odds ratios, and *p*-value from conditional multiple logistic regression after adjusting for sex, age of the child, residence, mother’s education, household expenditure, TV ownership, and calorie intake. FV is fruit and vegetables. SB is sedentary behavior.

**Figure 2 nutrients-12-00175-f002:**
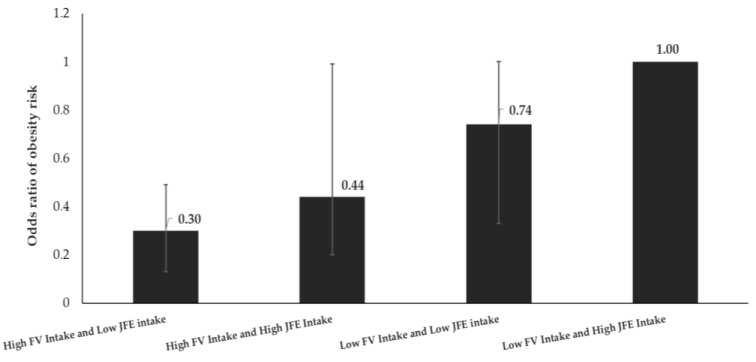
The interaction between fruit and vegetable intake and junk food energy intake. We generated the odds ratios and *p*-value from conditional multiple logistic regression after adjusting for sex, age of the child, residence, mother’s education, household expenditure, and TV ownership. FV is fruit and vegetable. JFE is junk food energy.

**Table 1 nutrients-12-00175-t001:** Characteristics of obese cases (*n* = 244) and non-obese controls (*n* = 244) 7–12 years of age, Yogyakarta, Indonesia.

	Obese ^a^	%/Mean ± SE	Non-Obese ^b^	%/Mean ± SE	χ^2^/t	*p*-Value
Gender						
Females	90	42.5	122	57.5	8.54	0.003
Males	154	55.8	122	44.2		
Children’s age						
≥10 years	106	49.5	108	50.5	0.03	0.855
<10 years	138	50.4	136	49.6		
School location						
Bantul	78	50.0	78	50.0	0.00	1
Yogyakarta	166	50.0	166	50.0		
Mother’s age						
≥40 years	98	51.6	92	48.4	0.31	0.578
<40 years	146	49.0	152	51.0		
Mother’s education						
Elementary school	13	41.9	18	58.1	1.58	0.663
Junior high school	31	47.0	35	53.0		
Senior high school	101	49.8	102	50.2		
University	99	52.7	89	47.3		
Mother’s job						
Unemployment	101	47.6	111	52.4	10.34	0.066
Farm workers	10	43.5	13	56.5		
Private employee	31	49.2	32	50.8		
Government employee	30	68.2	14	31.8		
Entrepreneur	64	52.9	57	47.1		
Others	8	32.0	17	68.0		
Father’s age						
≥40 years	142	49.8	143	50.2	0.01	0.927
<40 years	102	50.2	101	49.8		
Father’s education						
Elementary school	11	50.0	11	50.0	1.41	0.703
Junior high school	25	43.1	33	56.9		
Senior high school	98	50.0	98	50.0		
University	108	51.9	100	48.1		
TV ownership						
None or 1 TV	108	46.0	127	54.0	2.96	0.085
>1 TV	136	53.8	117	46.2		
Household Monthly Expenditure (Rp)	244	2,519,699 ± 96,892	244	2,611,081 ± 158,086	0.50	0.622
Mean of sedentary time (hours/d)	244	5.3 ± 0.052	244	4.4 ± 0.06	−10.8	<0.001
Mean of Fruit-and-Vegetable intake (serving/d)	244	2.39 ± 0.12	244	3.35 ± 0.17	4.6	<0.001
Mean Fast Food Energy Intake (kcal/d)	169	155.2 ± 14.6	161	112.4 ± 11.0	−2.3	0.0210
Mean Junk Food Energy Intake (kcal/d)	244	821.2 ± 32.7	244	702.6 ± 23.4	−2.9	0.0034

^a^ Obese children were those who have a body mass index (BMI) ≥95th percentile age and sex-specific body mass index proposed by the Centers for Disease Control and Prevention. ^b^ Non-Obese children were those who have BMI below 85th percentile age and sex-specific body mass index proposed by the Centers for Disease Control and Prevention.

**Table 2 nutrients-12-00175-t002:** Fruits & vegetables consumption, junk food energy intake, and sedentary behavior as risk factors of childhood obesity.

Risk Factors		Obese *n* (%)	Non-Obese *n* (%)	Crude OR ^a^ (95% CI)	Adjusted OR ^b^ (95% CI)	Adjusted OR ^c^ (95% CI)
Fruit & Vegetable intake as dichotomous variable	<3 serving/d	155 (57.0)	117 (43.0)	Ref	Ref	
	≥3 serving/d	89 (41.2)	127 (58.8)	0.53 (0.37–0.76)	0.46 (0.30–0.69)	
Junk Food Energy (JFE) Intake as a dichotomous variable	≥1050 kcal/d	66 (58.9)	46 (41.1)	Ref	Ref	
	<1050 kcal/d	178 (47.3)	198 (52.7)	0.63 (0.41–0.96)	0.61 (0.37–0.98)	
Sedentary Behavior as a dichotomous variable	≥5 h/d	155 (71.4)	62 (28.6)	Ref	Ref	
	<5 h/d	89 (32.8)	182 (67.2)	0.20 (0.13–0.29)	0.18 (0.12–0.28)	
Fruit & Vegetable Intake	Never/d	40 (69.0)	18 (31.0)	Ref		Ref
	1 serving/d	52 (62.7)	31 (37.4)	0.75 (0.37–1.53)		0.72 (0.32–1.60)
	2 serving/d	63 (48.1)	68 (51.9)	0.42 (0.21–0.80)		0.36 (0.17–0.77)
	≥3 serving/d	89 (41.2)	127 (58.8)	0.31 (0.17–0.59)		0.25 (0.12–0.50)
Junk Food Energy (JFE) Intake	High (≥1050 kcal/d)	66 (58.9)	46 (41.1)	Ref		Ref
	Moderate (700–1049 kcal/d)	57 (53.8)	49 (46.2)	0.81 (0.47–1.39)		0.69 (0.37–1.28)
	Low (<700 kcal/d)	121 (44.8)	149 (55.2)	0.57 (036–0.88)		0.53 (0.28–0.99)
Sedentary Behavior	High (≥ 5 h/d)	155 (71.4)	62 (28.6)	Ref		Ref
	Moderate (3.75–<5 h/d)	77 (35.8)	138 (64.2)	0.22 (0.15–0.33)		0.19 (0.12–0.30)
	Mild (<3.75 h/d)	12 (21.4)	44 (78.6)	0.10 (0.05–0.22)		0.09 (0.04–0.19)

^a^ Crude OR generated from a simple logistic regression model. ^b^ Adjusted OR derived from conditional multiple logistic regression model adjusting for sex, age of the child, residence, mother’s education, household expenditure, and TV ownership in which we treated fruit and vegetable consumption, junk food energy intake, and sedentary behavior as dichotomous variables. ^c^ Adjusted OR derived from conditional multiple logistic regression model adjusting for sex, age of the child, residence, mother’s education, household expenditure, and TV ownership in which we treated fruit and vegetable consumption, junk food energy intake, and sedentary behavior as categorical variables with three or more strata.

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
