# Peer review of "Improved Understanding of Interactions between Risk Factors for Child Obesity May Lead to Better Designed Prevention Policies and Programs in Indonesia"

_nutrients, 2020, doi:10.3390/nu12010175_

Round 1

Reviewer 1 Report

The authors significantly improved the manuscript and addressed most of my previous comments (I still do no agree with reporting the figures but it’s their choice). It is very appropriate they reported the instrument used, clarifying some issues about the material and methods. The discussion still report minor off-topic perspective but I am generally satisfied on the content.

Reviewer 2 Report

Manuscript Number Nutrients-684007, titled:  Improved Understanding of Interactions between Risk Factors for Child Obesity May Lead to Better Designed Prevention Policies and Programs in Indonesia

Review 2 –  December 19th, 2019

Dear Editor of Nutrients,

the authors have not completed the corrections I suggested in my rev-1.

In my opinion this manuscript presents scientific lacks and cannot be published in the present form. Please, re-send the authors my rev-1 to complete (100%) the corrections.

In addition they have to color in blue all the corrections they have made (rev1 and rev-2).

Best regards.

Round 2

Reviewer 2 Report

Manuscript Number Nutrients-684007, titled: Improved Understanding of Interactions between Risk Factors for Child Obesity May Lead to Better Designed Prevention Policies and Programs in Indonesia

Review 3 –  December 26th, 2019

Dear Editor of Nutrients,      

the authors have completed the corrections I suggested in my rev-1 and rev-2.

I suggest the publication of this manuscript as it is.

Best regards.

This manuscript is a resubmission of an earlier submission. The following is a list of the peer review reports and author responses from that submission.

Round 1

Reviewer 1 Report

Comments to the Authors:

Manuscript ID nutrients-634456

Type: Article

Title: Combinations of Fruit-and-Vegetable and Junk Food Intakes and Sedentary Behavior May Affect Risk of Child Obesity in Indonesia

Overview and general recommendations:

Dear authors,
first, thank you for the opportunity to review this manuscript which describes the

risk of Obesity in Indonesia taking into considerations some factors such as Fruit-and-Vegetable, Junk Food Intakes and Sedentary Behavior, as well as their associations.

The manuscript deals with a timely topic for developing countries (childhood obesity) which is appropriate for the Journal scope and area of interest.

However, the authors need to make clearer the originality of this study. There are other research studies which show correlations between the factors highlighted in this work. The novelty of the approach needs further investigation.

My suggestion is to improve the quality of the manuscript, especially the discussion, before further considering it for publication.

See the comments in the file.

Author Response

Dear The Reviewer 1,

Best regards,

Hamam Hadi

Reviewer 2 Report

Manuscript Number Nutrients-634456, titled: Combinations of Fruit-and-Vegetable and Junk Food Intakes and Sedentary Behavior May Affect Risk of Child Obesity in Indonesia

Review 1 –  27 November 2019

Dear Editor of Nutrients   

in general:

the argument is interesting and the experiment is well designed. The introduction has to be improved and extended. Some inaccuracies in the text. The references section has to be arranged using the guidelines of Nutrients.

In detail:

Abstract section, you have included and explained the abbreviations in the Introduction section, so there is no reason to duplicate this in the abstract section. Do not include the abbreviations in the abstract section; Line 53, insert a space after Health; Line 54, insert a space after day; Line 60, include the references 12, 13 and 14 between the same couple of brackets; Line 61, insert a space after children; Line 62, insert a space after obesity; Line 65, remove the dot after weight and place it after (18); Line 66, insert a space after balance; Line 66, after your reference 20, extend this section and explain that flavonoid content in fruit and vegetable is influenced by fertilizers [x1], ripening stage of fruit [x2], storage method [x3], cooking method [x4].

[x1] Use of digestate as an alternative to mineral fertilizer: effects on growth and crop quality.
Panuccio M.R.; Papalia T.; Attinà E.; Giuffrè A.; Muscolo A.
Archives of Agronomy and Soil Science 65 (5) 700-711 (2019).
https://doi.org/10.1080/03650340.2018.1520980
[x2] Bergamot (Citrus bergamia, Risso): The Effects of cultivar and harvest date on functional properties of juice and cloudy juice.
Giuffrè A.M.
Antioxidants 8, 221 (2019). doi:10.3390/antiox8070221
[x3] Physico-chemical Stability of Blood Orange Juice during Frozen Storage.
Giuffrè A.M., Zappia C., Capocasale M.
International Journal of Food Properties 20:sup2, 1930-1943 (2017).   
https://doi.org/10.1080/10942912.2017.1359184
[x4] Various cooking methods and the flavonoid content in onion.
Ioku K, Aoyama Y, Tokuno A, Terao J, Nakatani N, Takei Y.
J Nutr Sci Vitaminol (Tokyo). 2001, 47(1):78-83.

Line 72, include the references 23, 24 and 25 between the same couple of brackets; 1 section, explain why you have conducted this study in Bantul Regency and Yogyakarta City; Line 91, insert a space after … method; Line 109, insert a space after obesity; Line 119, include the references 30-31 between the same couple of brackets; Line 127, include the references 32-33 between the same couple of brackets; Tables 1 and 2, when you indicate (hours/ d), (kcal/ d), sometime you insert a space before d and sometime not. Please, always delete the space before d; Line 183, change servings/ d as servings/d; Line 186, change servings/ d as servings/d; Figure 2, in the y axis replace the comma with a dot, the same above the bars; Line 241, include the references 36-37 between the same couple of brackets; Line 241, include the references 39-40 between the same couple of brackets; Lines 251 and 252 include the references using the guidelines of Nutrients; Line 269, insert a space after decades, Line 271, insert a space after risk; Line 271, include the references using the guidelines of Nutrients; Conclusions not Conclusions. You have to extend this section with the most important results of your study; The References section has to be arranged using the guidelines of Nutrients and the citations have to be completed including volume number and page numbers. Please color in blue the corrections you will do.

I suggest a major revision

Best regards.

Author Response

Dear the Reviewer,

Best regards,

Hamam Hadi
